# Bandit Limited Discrepancy Search and Application to Machine Learning Pipeline Optimization

**Akihiro Kishimoto, Djallel Bouneffouf, Radu Marinescu, Parikshit Ram, Ambrish Rawat, Martin Wistuba,**[*] **Paulito Palmes, and Adi Botea**[†]
*IBM Research*

## Abstract

Optimizing a machine learning (ML) pipeline has been an important topic of AI and ML. Despite recent progress, this topic remains a challenging problem, due to potentially many combinations to consider as well as slow training and validation. We present the BLDS algorithm for optimized algorithm selection (ML operations) in a fixed ML pipeline structure. BLDS performs multi-fidelity optimization for selecting ML algorithms trained with smaller computational overhead, while controlling its pipeline search based on multi-armed bandit and limited discrepancy search. Our experiments on well-known benchmarks show that BLDS is superior to competing algorithms.

## 1. Introduction

Automated Machine Learning (AutoML) seeks to automatically compose and parameterize ML algorithms to maximize a given metric such as predictive accuracy on a given dataset. The task has become more acute in light of the recent explosion in ML applications. AutoML gradually extended from hyper-parameter optimization (HPO) for the best configuration of a single ML algorithm (Bergstra et al., 2011) to tackling the optimization of the entire ML pipeline from data preparation to model learning (Feurer et al., 2015). Consequently, this effort has spurred the development of many systems such as (Kotthoff et al., 2017; Olson et al., 2016; Feurer et al., 2015; Mohr et al., 2018).

Existing AutoML systems often assume a fixed linear structure of the pipeline consisting of several different stages such as pre-processing, feature selection, transformation and estimation. The AutoML problem known as the *combined algorithm selection and hyper-parameter optimization (CASH)* selects the ML methods for each of these pipeline stages and the corresponding hyper-parameters of these methods such that a given black-box objective function is optimized. More specifically, the fixed pipeline structure implies an optimization problem with a fixed number of decision variables where, for example, we have one variable for a preprocessing algorithm, one variable for a learning algorithm, and one variable for each parameter of each algorithm. This in turn leads to a complex solution space involving both discrete and continuous variables. Consequently, the CASH problem is solved in many different ways, including standard Bayesian optimization (BO) (Hutter et al., 2011; Klein et al., 2017; Falkner et al., 2018), hierarchical task networks (Mohr et al., 2018; Katz et al., 2020), reinforcement learning (Drori et al., 2018) or Monte-Carlo tree search coupled with BO (Rakotoarison et al., 2019). There are other AutoML tasks including pipeline generations (Olson et al., 2016; Marinescu et al., 2021) but they are beyond the scope of this paper.

A recent approach to tackle CASH that proved quite effective in practice splits the algorithm selection phase and the HPO into two simpler subproblems which are subsequently solved separately

---

∗. Currently at Amazon Research, Germany. Work performed while the author was affiliated with IBM Research.
†. Currently at Eaton, Ireland. Work performed while the author was affiliated with IBM Research.

in an iterative manner using the augmented Lagrangian function via the alternating direction method of multipliers (ADMM) (Liu et al., 2020). This general framework was shown to outperform existing AutoML systems that solve CASH as a full joint optimization problem, e.g., (Feurer et al., 2015; Olson et al., 2016). However, even when solved in isolation, the algorithm selection subproblem raises the following challenges: (1) the black-box nature of the objective function prevents the algorithm selection from leveraging any of the objective function's characteristics while searching for a better pipeline configuration; (2) there are many possible combinations of algorithms in a multi-stage pipeline structure and training every single pipeline configuration is actually quite expensive especially when dealing with large input datasets.

**Contribution:** We focus on the *algorithm selection* problem in AutoML and introduce a new algorithm, called Bandit Limited Discrepancy Search (BLDS), that combines ideas behind the multi-armed bandit (MAB) algorithms, e.g., (Auer et al., 2002; Kocsis and Szepesvári, 2006), limited discrepancy search (LDS) (Harvey and Ginsberg, 1995) and multi-fidelity optimization (Sabharwal et al., 2016). More specifically, BLDS assumes that a better solution tends to be found in a set of pipelines similar to the current best one. The notion of discrepancy in LDS reduces the search space examined by BLDS. In addition, BLDS attempts to reduce the computational overhead associated with training the pipelines. It starts with a small subset of training data and increases the size of the subset if that pipeline is promising according to an objective function. This involves a procedure to compare pipelines trained with different subsets of data. Unlike existing multi-fidelity optimization approaches such as the Data Allocation Upper Bound (DAUB) algorithm (Sabharwal et al., 2016), BLDS calculates the upper and lower confidence bounds inspired by MAB. These bounds allow BLDS to select a more promising pipeline as well as to decide whether to allocate more resources to a pipeline for further training. We compare BLDS with the Combinatorial MAB (CMAB) algorithm (Liu et al., 2020) as well as DAUB and Hyperband (Li et al., 2018), which we adapted for algorithm selection. Using well-known ML benchmarks and a fixed 4-stage pipeline structure comprising over 3000 possible algorithm selections, we show that BLDS performs better than the competing methods.

The appendix includes details of the algorithms, empirical evaluation and additional results.

## 2. Preliminaries

**Algorithm Selection** An ML pipeline structure consists of a fixed sequence of $m$ stages (e.g., data preprocessor $\rightarrow$ feature preprocessor $\rightarrow$ classifier) such that for each stage $j = 1, \ldots m$ a set $\mathcal{A}_j$ of ML algorithms is available. A pipeline configuration (or pipeline for short) $p \in \mathcal{P}$ is a complete configuration of algorithms, one for each stage, namely $p = (a_1, \ldots, a_m)$ where $a_j \in \mathcal{A}_j$ is the algorithm selected for the $j$-th stage and $\mathcal{P} = \mathcal{A}_1 \times \cdots \times \mathcal{A}_m$ is the set of all possible pipeline configurations. Given a limited amount of training data $\mathcal{D} = \{(\mathbf{x}_1, y_1), \ldots, (\mathbf{x}_n, y_n)\}$, the goal of the algorithm selection problem is to determine the pipeline $P^* \in \mathcal{P}$ with optimal generalization performance estimated by splitting $\mathcal{D}$ into disjoint training and validation sets $\mathcal{D}_{train}^{(i)}$ and $\mathcal{D}_{valid}^{(i)}$, learning functions $f_i$ by applying $P^*$ to $\mathcal{D}_{train}^{(i)}$ and evaluating the predictive performance of these functions on $\mathcal{D}_{valid}^{(i)}$. More formally, the problem is written as $p^* = \arg\min_{p \in \mathcal{P}} \frac{1}{k} \sum_{i=1}^{k} \mathcal{L}(p, \mathcal{D}_{train}^{(i)}, \mathcal{D}_{valid}^{(i)})$, where $\mathcal{L}(p, \mathcal{D}_{train}^{(i)}, \mathcal{D}_{valid}^{(i)})$ is the value of the black-box objective function (e.g., misclassification error) achieved by $p$ when trained on $\mathcal{D}_{train}^{(i)}$ and evaluated on $\mathcal{D}_{valid}^{(i)}$.

---

**Algorithm 1** BLDS: Bandit Limited Discrepancy Search

---

**Require:** Training set $T$, validation set $V$, discrepancy $disc$
1: **procedure** BLDS($T, V, disc$)
2:  **while** time is not up **do**
3:   $p_{init}$ = FINDINITIALPIPELINE()
4:   **repeat**
5:    INCREASEANDTRAIN($p_{init}, T, V$)
6:    **for all** ($\theta = 1; \theta \leq disc; \theta = \theta + 1$) **do**
7:     $p_{new}$ = SEARCH($p_{init}, p_{init}.lcb, p_{init}.ucb, T, V, 1, \theta$)
8:     **if** ($p_{new} \neq \phi$) **then**
9:      $p_{init} = p_{new}$; **break**
10:   **until** ($p_{init}$ is trained with a full set of $T$)
11:  **return** best pipeline obtained
12: **function** SEARCH(Pipeline $p$, LCB $lcb$, UCB $ucb$, training set $T$, validation set $V$, stage $i$, current discrepancy $\theta$)

13:  **if** ($\theta = 0 \vee i > m$) **then**
14:   $(l, u)$ = GETCURRENTPERFORMANCE($p$)
15:   **if** ($u < lcb$) **then return** $p$
16:   **if** ($l \leq ucb$) **then** $(l, u)$ = INCREASEANDTRAIN($p, T, V$)
17:   **if** ($u < ucb$) **then return** $p$
18:   **else return** $\phi$
19:  **else**
20:   **for all** algorithm $a \in \mathcal{A}_i$ **do**
21:    **if** ($p[i] = a$) **then**
22:     $r$ = SEARCH($p, lcb, ucb, T, V, i + 1, \theta$)
23:    **else**
24:     $p_{new} = p$; $p_{new}[i] = a$
25:     $r$ = SEARCH($p_{new}, lcb, ucb, T, V, i + 1, \theta - 1$)
26:    **if** ($r \neq \phi$) **then return** $r$
27:   **return** $\phi$

---

**Limited Discrepancy Search** Assume that a search space is a *complete binary tree* with height $h$. Leaf nodes correspond to *goals* or *failures* and the task of interest is to find a goal leaf node. Each internal node represents a decision that has to be made to reach a goal. Furthermore, the left child of each internal node represents following the recommendation of a value-ordering heuristic and the right child represents going against that recommendation. Disregarding the recommendation is called a *discrepancy* (Harvey and Ginsberg, 1995). The number of discrepancies of a leaf node is the number of right turns in the path from the root to that leaf node. *Limited Discrepancy Search* (LDS) (Harvey and Ginsberg, 1995; Korf, 1996) is a depth-first search algorithm that searches for a goal node while increasing the number of discrepancies in an iterative manner (see Appendix for details).

## 3. Bandit Limited Discrepancy Search

In this section, we present our new *Bandit Limited Discrepancy Search* (BLDS) algorithm for tackling the algorithm selection problem in AutoML. The basic idea behind our approach is to conduct a discrepancy-based exploration that focuses on the most promising portions of the pipeline search space while controlling the size of the training data used for training the pipelines found in a most cost-effective manner. Unlike standard LDS, we consider an optimization problem where each leaf node is a goal and corresponds to a pipeline $p$ which has an associated cost (i.e., value of a black-box loss or objective function $\mathcal{L}(p)$) and the task is to find the least-cost one.

### 3.1 Algorithm Description

Algorithm 1 describes the BLDS approach. We consider a linear[1] pipeline structure with $m$ stages such that each stage $i$ has a set $\mathcal{A}_i$ of available machine learning algorithms. The following notations are used. Function FINDINITIALPIPELINE generates a randomly initialized pipeline $p_{init} = (a_1, \ldots, a_m)$, however, the algorithm can start with any pipeline obtained with other Au-

---

1. BLDS is applicable to non-linear pipeline structures as well.

toML methods. Function INCREASEANDTRAIN trains pipeline $p$ with a (sub)set of training data $T$ and evaluates $p$'s performance on validation data $V$. As also discussed in (Sabharwal et al., 2016), when training $p$ for the $k$-th time, BLDS selects $b\eta^k$ samples from $T$, where $b$ and $\eta$ are constants. The pipeline $p$ has an extra structure that preserves an objective value $v$, and two associated values $lcb$ and $ucb$ which refer to a lower confidence bound (LCB) and an upper confidence bound (UCB), respectively. The LCB and UCB values are regarded as lower and upper bounds of the achievable objective value (we defer the details to the next subsection). The discrepancy $disc$ indicates the maximum number of allowed algorithm changes to the stages of the initial pipeline $p_{init}$. BLDS assumes that a better pipeline tends to be instantiated in a similar fashion to $p_{init}$ and, therefore, it examines a limited search space where similar pipelines are located. The symbol $\phi$ is used to indicate that the algorithm could not find a pipeline better than $p_{init}$ with current discrepancy value $\theta$.

The algorithm starts with a discrepancy value $\theta$ of 1 and conducts an iterative search that allows to change the algorithms of at most $\theta$ stages in $p_{init}$ while incrementing $\theta$ until a better pipeline $p_{new}$ is found or $\theta$ exceeds $disc$ (see lines 5-10). If $p_{new}$ is found, BLDS uses it as a pipeline to attempt to improve further, starting with $\theta = 1$. BLDS repeats the steps of INCREASEANDTRAIN and the iterative search limited by $\theta$ until it finds a pipeline trained with a full set $T$. However, even after finding such a pipeline, the algorithm can still continue the search for another one by restarting with a different initial pipeline calculated by FINDINITIALPIPELINE until it uses up the allocated time. For efficiency, BLDS also caches the objective value and the corresponding UCB/LCB values for all trained pipelines in order to avoid retraining them.

Function SEARCH (lines 12-27) performs the actual exploration of the pipeline search space limited by discrepancy $\theta$. Specifically, when it selects an algorithm $a$ that is different from the one corresponding to stage $i$ in $p_{init}$ it decrements $\theta$ to reduce the number of changes allowed for the remaining stages (lines 24-25). Otherwise, the algorithm for stage $i$ is unchanged and, therefore, the $\theta$ value is preserved (line 21-22). When SEARCH either has checked all $m$ stages in $p$ or consumed the discrepancy budget, it checks $p$'s performance (lines 13-18). The GETCURRENTPERFORMANCE method retrieves $p$'s LCB and UCB values $l$ and $u$ if they are cached. Caching alleviates the overhead of revisiting the same pipelines possibly with different $\theta$. Otherwise, it evaluates $p$ with $V$ after training $p$ with $b$ samples in $T$. BLDS assumes the real objective value for $p$ to be in $[l, u]$ and decides whether or not $p$ is a promising pipeline as well as whether or not $p$ should be trained with a larger training subset. This way, BLDS attempts to focus on promising pipelines and thus alleviates the training overhead. SEARCH receives $p_{init}$'s LCB and UCB values $lcb$ and $ucb$. If $u < lcb$ holds, $p$ is considered to be better than $p_{init}$ and becomes a new pipeline to start with (line 15). If $l \leq ucb$ (and $lcb \leq u$) holds, $p$ might or might not be better than $p_{init}$. In this case, $p$ is re-trained with an increased training (sub)set and re-evaluated with $V$. The algorithm subsequently selects a pipeline based on whether or not $p$'s updated UCB value is better than that of $p_{init}$ (lines 16-17). For the other cases (e.g., $ucb \leq l$ holds), $p_{init}$ is considered to be better than $p$ and thus is kept (line 18).

### 3.2 Upper and Lower Confidence Bounds

Existing approaches to compute UCB values for addressing the MAB problems only account for the number of visits to each arm/branch, e.g., (Auer et al., 2002; Kocsis and Szepesvári, 2006). In our task, as defined below, the size of the training (sub)set is closely related to the accuracy of the objective value. Therefore, in addition to the number of training operations performed, our UCB and LCB formulas newly account for the training data size: $UCB = v + \sqrt{\frac{\log \frac{cLD_k{}^2}{\delta}}{D_k}}$ and

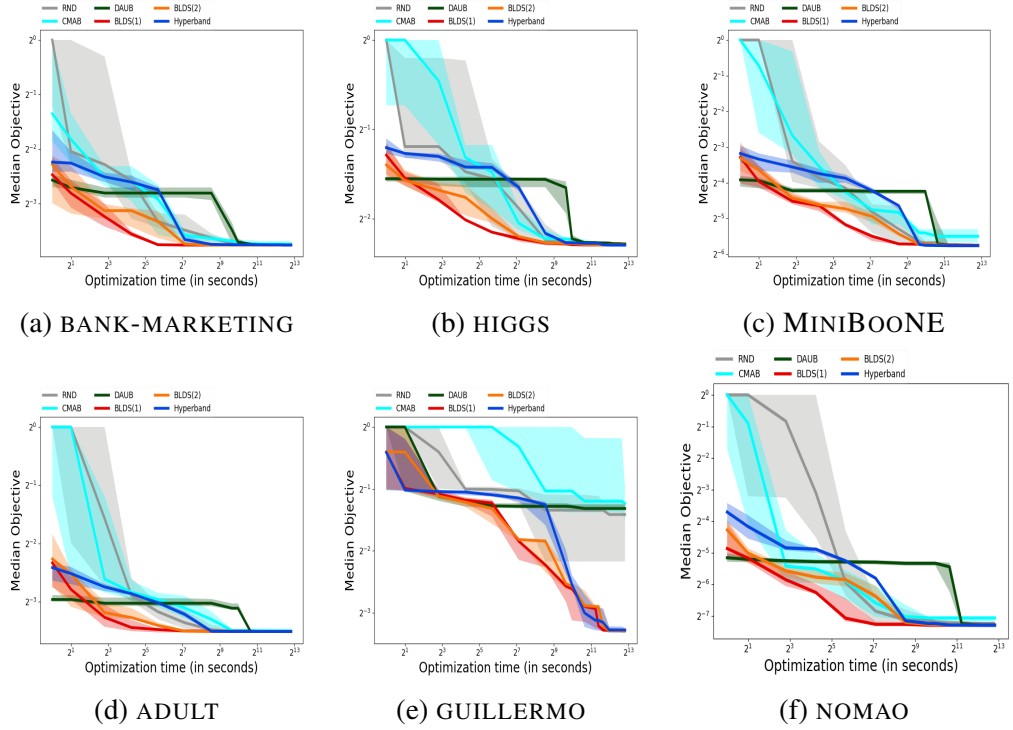

Figure 1: Performance of each method for algorithm selection for representative domains.

$LCB = v - \sqrt{\frac{\log \frac{cLD_k{}^2}{\delta}}{D_k}}$, where $v$ is an objective value for the $k$-th evaluation with validation set $V$ and $c$ $(> 4)$ and $\delta$ are constants, $L = \Pi_{i=1}^m |\mathcal{A}_i|$ and $D_k = \sum_{j=1}^k b\eta^{(j-1)}$.

The second term of UCB/LCB determines whether to perform a so-called *exploration*, aiming at updating $v$ that might be inaccurate due to a small number of training examples.

## 4. Experimental Results

We implemented all algorithms in Python using scikit-learn (Pedregosa et al., 2011) and performed the experiments on a cluster of Intel Xeon CPU E5-2667 processors at 3.3GHz. We use only one core when running each algorithm in order to better track the objective value versus time as also suggested in (Rakotoarison et al., 2019; Liu et al., 2020). We evaluate: (a) BLDS(1) and BLDS(2) with $disc = 1, 2$, respectively, (b) CMAB (Liu et al., 2020) (c) DAUB, (d) Hyperband and (e) simple random search (RND). Although algorithms DAUB and Hyperband were originally designed to address other tasks, we adapted them here to pipeline optimization (see the Appendix).

We set up experiments similar to those in (Liu et al., 2020). For each benchmark dataset, we consider a 70-30% train-validation split, and run each algorithm with a time limit of two hours per trial to perform a binary classification. We consider $(1.0 - \text{AUROC})$ (area under the ROC curve) as the black-box objective function that needs to be minimized. We select 10 benchmarks from OpenML repositories (Bischl et al., 2017). For a consistent evaluation, we first impute any missing values with the most common value of the corresponding feature and subsequently perform one-hot encoding of the categorical features. For our purpose, we consider a 4-stage pipeline structure which results

in a total of 3072 possible pipelines.[2] For each benchmark, all algorithms use the same training and validation sets. BLDS, Hyperband and DAUB use the same strategy to increase the subset of training data for their multi-fidelity optimization. We set $b = 100$ and $\eta = 2$ for all the algorithms and $cL/\delta = 1/9600$ for BLDS, where $L = 3072$.

Figures 1(a)-(f) show the performance for representative benchmarks. For clarity, we use doubly logarithmic plots. After 10 runs for each algorithm, we compute a median of the objective values and the region within the first and third quartiles. The fixed default parameters from scikit-learn are used for each instantiated pipeline and no HPO is performed.

These results clearly demonstrate that BLDS(1) tends to achieve better objective values much quicker than the other competitors. For example, BLDS(1) achieves the objective value of 0.07398 in 138 seconds on BANK-MARKETING, thus converging 19.4 times faster than Hyperband. Moreover, BLDS(1) tends to outperform the other methods for the first 30-500 seconds in many cases including datasets ADULT, BANK-MARKETING, and MINIBOONE. When running for a longer time, the other schemes are able to catch up with BLDS(1). At this stage, they can evaluate a sufficiently large number of pipelines, thus being able to return the objective values competitive to those found by BLDS(1). Liu et al. (2020) showed that CMAB is the best-performing algorithm under the ADMM framework. However, we observe that CMAB sometimes converges to values that are much worse than those obtained by the other methods. RND also occasionally suffers from a suboptimal solution. These results might indicate that the multi-fidelity optimization has an advantage over these approaches which are not based on multi-fidelity optimization. DAUB performs poorly in general. Due to more configurations (i.e., 3072 pipelines) than those in (Sabharwal et al., 2016) (only 41 ML classifiers to choose from), DAUB suffers from a significant overhead in its bootstrapping step. Even in its pipeline search step, DAUB's linear regression model is not often accurate enough to return an optimized pipeline. DAUB needs to continue search even after a fully-trained pipeline is obtained. DAUB eventually finds an optimized pipeline, but suffers from much slower convergence. BLDS(2) under-performs BLDS(1). We hypothesize that this is caused by a much larger number of pipelines needed to be re-trained and evaluated within a discrepancy threshold $disc$. If BLDS(1) finds no pipeline better than the initial one, it examines 26 pipelines within $disc = 1$. BLDS(1) then restarts with a new pipeline randomly initialized, which might be a good starting point. However, in the worst case, BLDS(2) needs to examine 272 pipelines within $disc = 2$, before it restarts search with a new initial pipeline. Therefore, there is a big gap in the local search space between $disc = 1$ and 2.

## 5. Conclusion

We introduced BLDS to address the algorithm selection problem in AutoML. Our results clearly show that BLDS performs well and tends to converge more quickly than other competing algorithms. In future work, we plan to further enhance search performance to be able to deal with large-scale training data as well as more complicated pipeline structures and combine BLDS with HPO under AutoML ADMM. Possible extensions include a combination with an approach for selecting candidate pipelines with meta-learning, e.g., (Feurer et al., 2015; Rakotoarison et al., 2019), an introduction of a better discrepancy value allowing for a more granular control of the local search space, and parallelization of BLDS. Applying BLDS to other tasks such as HPO is also important to elucidate its applicability and limitation. Finally, it is important to have a better theoretical understanding to our MAB strategy by taking into account BLDS' behavior that limits the search space.

---

2. See the supplementary material for the ML algorithms considered at each stage and the data size of each benchmark.

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

---
**Algorithm 2** LDS: Limited Discrepancy Search

---
1: **procedure** LDS
2:     **for all** $k = 0 \ldots n$ **do**
3:         **if** PROBE($root, k$) **then return** $true$
4: **function** PROBE($node, k$)
5:     **if** isLeaf($node$) **then return** isGoal($node$)
6:     **if** $k = 0$ **then return** PROBE(left(node), 0)
7:     **else return** PROBE(right($node$), $k$-1) or PROBE(left($node$),$k$)

---

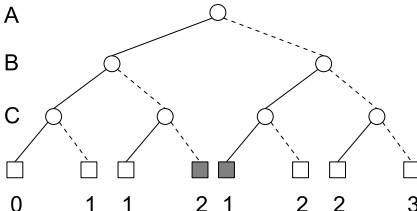

Figure 2: Search space traversed by LDS (the height is 3). The number of discrepancies is indicated below the leaf nodes.

## Appendix A. Limited Discrepancy Search

We consider a search space that is a *complete binary tree* with bounded height $h$. Leaf nodes correspond to *goals* or *failures* and the task of interest is to find a goal leaf node. Each internal node represents a decision that has to be made to reach a goal.

Furthermore, the left child of each internal node represents following the recommendation of a value-ordering heuristic and the right child represents going against that recommendation. Disregarding the heuristic recommendation is called a *discrepancy* (Harvey and Ginsberg, 1995). The number of discrepancies of a leaf node is the number of right turns in the path from the root to that leaf node.

*Limited Discrepancy Search* (LDS) (Harvey and Ginsberg, 1995; Korf, 1996) is a depth-first search algorithm that searches for a goal node while increasing the number of discrepancies in an iterative manner. The pseudo-code is given in Algorithm 2. The $k$-th iteration of the main loop will visit all the leaves having $k$ or fewer discrepancies. Function PROBE is a standard recursive implementation of depth-first search such that: (i) it keeps track (parameter $k$) of the number of discrepancies still available, (ii) if a discrepancy is consumed, $k$ is decreased before the recursive call and (iii) if no further discrepancies are available, the algorithm does not disregard the heuristic. Since the last iteration visits all the leaves, the algorithm is complete. In practice, LDS is used in a anytime manner until a solution is found or a time limit is reached.

**Example 1** *Figure 2 shows a search tree with height 3. The gray leaves correspond to goals (solutions). LDS stops during iteration $k = 1$ where it finds the solution with 1 discrepancy.*

LDS has also been used successfully in optimization problems, where each leaf has an associated cost and the task is to find the least-cost one. In this case, the heuristic gives advise about the successor having the lowest cost leaf below and LDS outputs upper bounds on the optimal solution cost that improve over time.

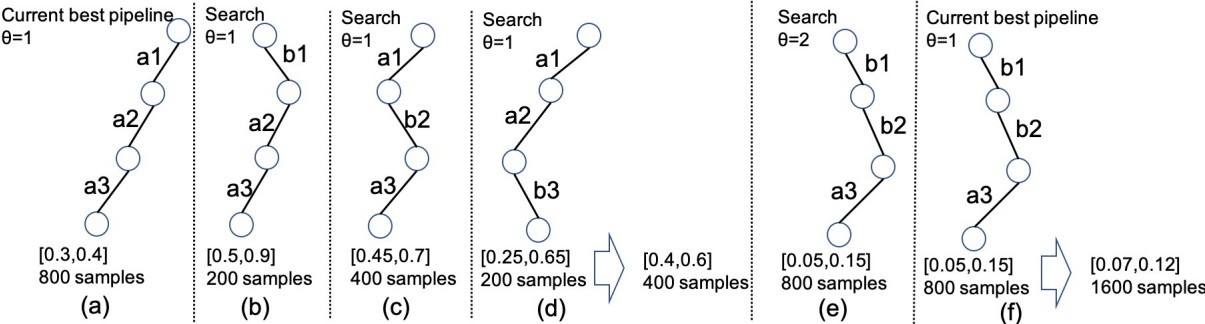

Figure 3: Search behavior of BLDS ($\eta = 2$) for a three-step pipeline structure where each module has two algorithm choices.

## Appendix B. Example of BLDS

Figure 3 illustrates the execution of BLDS on a 3-stage pipeline structure. The UCB and LCB values $u$ and $l$ of a pipeline are written as $[l, u]$. Let $p_1 = (a1, a2, a3)$ be the current best pipeline trained with 800 samples (Fig. 3(a)). BLDS limits search with $\theta = 1$, allowing to change only one stage in $p_1$. Figures 3(b)-(d) show the pipelines examined with $\theta = 1$. In Figure 3(b), BLDS examines pipeline $p_2 = (b1, a2, a3)$ but considers that $p_1$ is better than $p_2$, since $p_2$'s LCB value is larger than $p_1$'s UCB value. So is the case for pipeline $(a1, b2, a3)$ (Fig. 3(c)). In Figure 3(d), there is an overlap between the regions of the LCB and UCB values for pipelines $p_1$ and $p_3 = (a1, a2, b3)$. To obtain a more accurate objective value, BLDS re-trains $p_3$ with an increased training subset (i.e., 400 samples) and updates its LCB and UCB values. BLDS finds that $p_1$ is still better than $p_3$. Since BLDS cannot find a pipeline better than $p_1$ with $\theta = 1$, it sets $\theta = 2$, allowing to modify any of two modules in $p_1$. BLDS reaches pipeline $p_4 = (b1, b2, a3)$ (Fig. 3(e)). The UCB value of $p_4$ is smaller than $p_1$'s LCB value, indicating that $p_4$ is better than $p_1$. Therefore, BLDS stops searching with $\theta = 2$, sets $p_4$ to the new best pipeline and resets $\theta = 1$. By using $p_4$ as a new initial pipeline, BLDS re-trains $p_4$ with an increased training subset (i.e., 1600 samples) and obtains new LCB and UCB values. BLDS performs search with $\theta = 1$, allowing only one change to $p_4$.

## Appendix C. Extensions to DAUB and Hyperband

We adapt DAUB (Sabharwal et al., 2016) and Hyperband (Li et al., 2018) to obtain competing multi-fidelity optimization based baselines for the algorithm selection problem.

Let $T$ and $V$ be the training and validation sets. Our DAUB implementation generates all possible pipelines for a given fixed $m$-stage pipeline structure, and performs the steps of Sabharwal et al. (2016). Even if DAUB returns a first pipeline $p$ trained with full $T$, it can continue to run to return a second pipeline which is trained with full $T$ and which might perform better than $p$ with respect to an objective value evaluated with respect to $V$. Until DAUB's priority queue becomes empty, DAUB can keep searching for a better pipeline in this way. In a combination with HPO under the ADMM framework (Liu et al., 2020), we define the number of DAUB's iterations as the number of pipelines with full $T$ returned by DAUB. The pipelines not selected by DAUB are enqueued back to its priority queue. The selected pipeline is enqueued after its hyperparameters are optimized by an HPO algorithm.

Table 1: Machine learning algorithms used in the 4-stage pipeline structure. None indicates that no algorithm is selected.

| Step | Module |
|---|---|
| 1 (8 scalers) | None, Normalizer, Quantile transformer, Binarizer, Standard scaler, Robust scaler, MinMax scaler, KBins discretizer (ordinal encoding) |
| 2 (8 transformers) | None, Sparse random projection (dense output), Gaussian random projection, RBF sampler, PCA, Fast ICA, Truncated SVD (algorithm=randomized), Factor analysis (SVD method=randomized) |
| 3 (6 selectors) | None, Select percentile, Select Fpr, Select Fdr, Select FweFS, Variance threshold |
| 4 (8 estimators) | Random Forest, Logistic regression, Gaussian NB, KNeighbors, Quadratic discriminant analysis, AdaBoost (base estimator=decision tree, max depth=3), Extra trees, Decision tree |

Table 2: Data size of each benchmark

| | ADULT | JM1 | BANK MARKETING | NOMAO | AMAZON EMPLOYEE ACCESS | HIGGS | APSFAILURE | MINIBOONE | GUILLERMO | RICCARDO |
|---|---|---|---|---|---|---|---|---|---|---|
| size | 48,842 | 10,885 | 45,211 | 34,465 | 32,769 | 98,050 | 76,000 | 130,064 | 20,000 | 20,000 |

Our Hyperband implementation generates pipelines by random sampling and optimizes them by the steps of Li et al. (2018), which we define as one iteration. In case of searching for pipelines without HPO, this iteration can be repeated until the time is up. We set the maximum amount of resource (called parameter $R$) to the training data size. This allows to perform an increase of the training data subset in a similar way to DAUB and BLDS, while controlling the size of selected pipelines based on $\eta$. It also employs a caching scheme similar to that of BLDS, which effectively reuses previously trained pipelines within its Hyperband runs and among different Hyperband runs with a combination of HPO under the ADMM framework (Liu et al., 2020).

## Appendix D. Experimental Results

Figure 1 shows the performance of each algorithm selection method for all benchmarks. Table 1 shows the ML algorithms considered at each stage in the four-stage pipeline. Table 2 shows the data size of each benchmark.

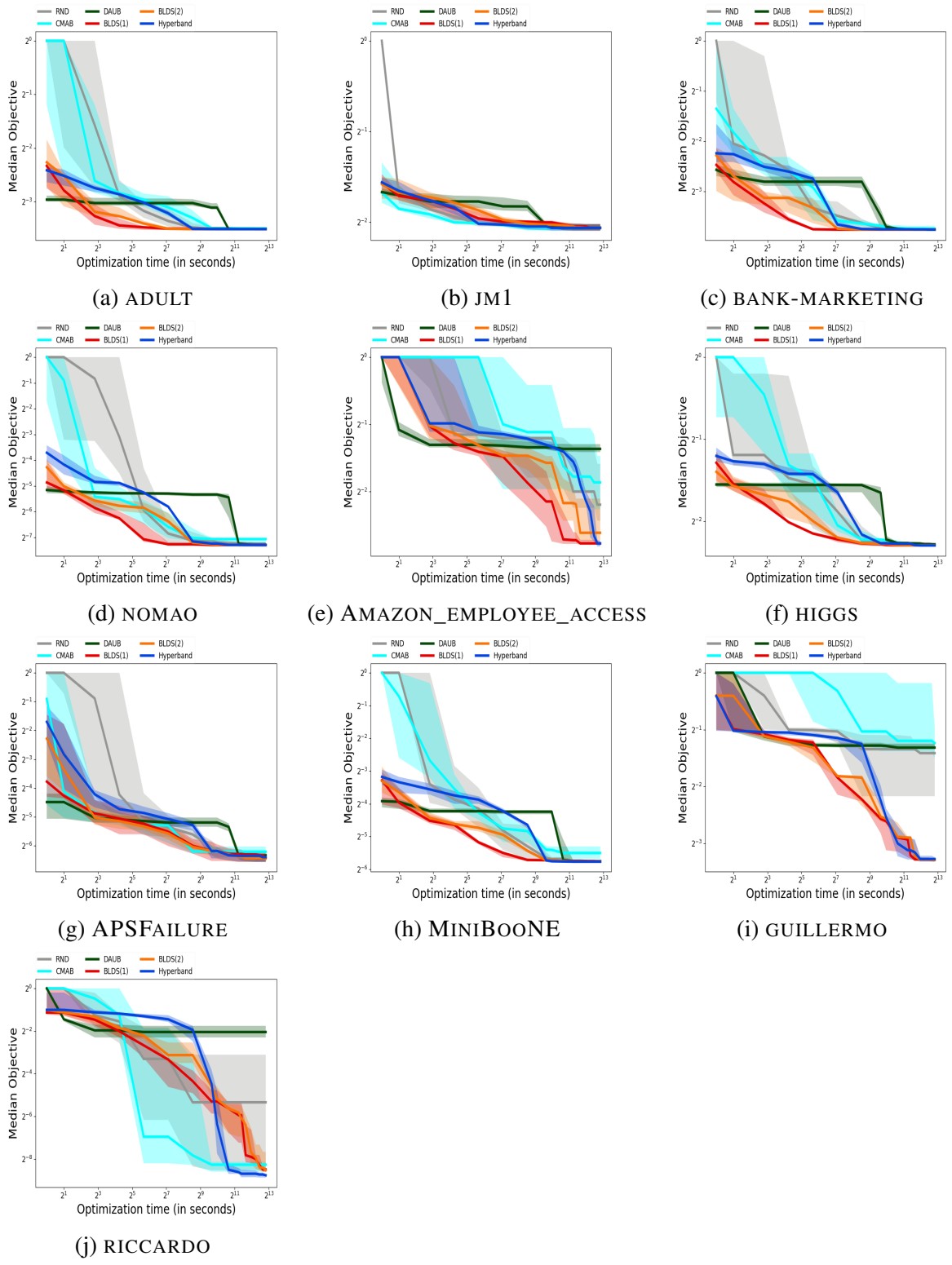

Figure 4: Performance of each method for algorithm selection

