# OpenReview forum: "Bandit Limited Discrepancy Search and Application to Machine Learning Pipeline Optimization"
_ICML.cc/2021/Workshop/AutoML — AutoML@ICML2021 Poster_

### Official Review · Reviewer_UJTy · 2021-06-13
**Good paper with interesting ideas**

**Rating:** 8
**Confidence:** 4

**Review:**

This paper proposes a Bandit Limited Discrepancy Search algorithm that also leverages the multi-fidelity optimization to treat the algorithm selection problem. The LDS part is used to reduce the search space as it is assumed that a better solution is not far from the current one. The algorithm then uses also bandit-based multi-fidelity technique to gradually assign resources to more promising pipeline.

The paper is sound and well written and the proposed idea is very interesting in the sense that it leverages the LDS to reduce the search space with a quite plausible assumption (that a better solution is often close to the current best one). Experimental results are also quite convincing. Otherwise, in addition to the present experiments, it would probably be interesting to add some ablation study, in particular to see what is the real impact of LDS (without multi-fidelity optimization on the resources).

Overall, this is a good paper for me for the workshop and I clearly recommend accept.

---

### Official Review · Reviewer_4c3U · 2021-06-15
**Algorithm Selection for ML Pipelines via BLDS**

**Rating:** 7
**Confidence:** 5

**Review:**

# Summary
In the paper "Bandit Limited Discrepancy Search and Application to Machine Learning Pipeline Optimization" the authors propose a bandit algorithm combined with a limited discrepancy search to optimize the algorithm choices in a four-step pipeline, consisting of feature scalers, transformers, selectors, as well as estimators.

# Comments

Existing AutoML systems assume a fixed linear structure of the pipeline consisting of several different stages such as pre-processing, feature selection, transformation and estimation."
This is not true at all. There are indeed AutoML systems that optimize for more complex pipelines shapes:
Olson, Randal S., et al. "Evaluation of a tree-based pipeline optimization tool for automating data science." Proceedings of the Genetic and Evolutionary Computation Conference 2016. 2016.
Wever, Marcel Dominik, Felix Mohr, and Eyke Hüllermeier. "Ml-plan for unlimited-length machine learning pipelines." ICML 2018 AutoML Workshop. 2018.
Marinescu, Radu, et al. "Searching for Machine Learning Pipelines Using a Context-Free Grammar." Proceedings of the AAAI Conference on Artificial Intelligence. Vol. 35. No. 10. 2021.

In the introduction the citation of Drori et al. (2018) is not in parentheses.

The authors state that "popular black-box solvers such as BO perform well with continuous variables but ar still not well established for combinatorial problems." However, in the experiments BLDS is not compared to BO at all, leaving the statement without any proof. Either the authors should provide a reference which gives evidence or prove it themselves via a proper empirical evaluation. Also a comparison to BOHB, a hybrid of BO and Hyperband, would be interesting:
Falkner, Stefan, Aaron Klein, and Frank Hutter. "BOHB: Robust and efficient hyperparameter optimization at scale." International Conference on Machine Learning. PMLR, 2018.

p. 3: What is a "black-box loss"?

Speaking about proper empirical evaluation, the number of datasets is comparatively low, even for a workshop paper. Also the choice of datasets appears to be rather arbitrary. Most of the datasets that have been used for benchmarking AutoML systems previously are not used at all, making it hard to relate the results to previous studies.

While for benchmarking considering non-parallel optimization is fine, parallelization is an important speed up to deploy AutoML systems in practice. How well can the BLDS search be parallelized? What are limitations of the approach?

3000 pipelines are not that much - so one could exhaustively search the space of different pipelines. To what extent are the approaches capable of finding the optimum?


# Language
p. 2: "A ML pipeline structure" => An ML pipeline structure
p. 3: "The following notations is used." => The following notation is used.
p. 4: "[...], BLDS uses it as new initial pipeline." => This might be confusing. How can there be a "new initial" pipeline? There can only be one *initial* pipeline.
p. 5: "[...], we consider an 70-30% train-validation split" => a 70-30% train-validation split
p. 6: "[...] large-scale data as well as more complicated pipelines structures as well as combine BLDS with HPO [...]" => two times "as well as"

---

### Decision · Program_Chairs · 2021-06-21

Accept (Poster)